# Randomised controlled trial of population screening for atrial fibrillation in people aged 70 years and over to reduce stroke: protocol for the SAFER trial

Jonathan Mant,[1] Rakesh N Modi [iD],[1] Andrew Dymond,[1] Natalie Armstrong [iD],[2] Jenni Burt,[3] Peter Calvert,[4] Martin Cowie,[5] Wern Yew Ding,[4] Duncan Edwards,[1] Ben Freedman,[6] Simon J Griffin,[7,8] Sarah Hoare,[9] F D Richard Hobbs,[10] Rachel Johnson,[11] Stephen Kaptoge,[12] Gregory Y H Lip [iD],[4,13] Trudie Lobban,[14] Mark Lown [iD],[15] Jenny Lund,[16] Richard J McManus [iD],[10] Mark T Mills,[4] Stephen Morris [iD],[9] Alison Powell,[17] Riccardo Proietti,[4] Stephen Sutton [iD],[9] Mike Sweeting,[12] Howard Thom,[11] Kate Williams [iD],[1] The SAFER author group

For numbered affiliations see end of article.

**Correspondence to**
Dr Rakesh N Modi;
rnm30@medschl.cam.ac.uk

## ABSTRACT

**Introduction** There is a lack of evidence that the benefits of screening for atrial fibrillation (AF) outweigh the harms. Following the completion of the Screening for Atrial Fibrillation with ECG to Reduce stroke (SAFER) pilot trial, the aim of the main SAFER trial is to establish whether population screening for AF reduces incidence of stroke risk.

**Methods and analysis** Approximately 82 000 people aged 70 years and over and not on oral anticoagulation are being recruited from general practices in England. Patients on the palliative care register or residents in a nursing home are excluded. Eligible people are identified using electronic patient records from general practices and sent an invitation and consent form to participate by post. Consenting participants are randomised at a ratio of 2:1 (control:intervention) with clustering by household. Those randomised to the intervention arm are sent an information leaflet inviting them to participate in screening, which involves use of a handheld single-lead ECG four times a day for 3 weeks. ECG traces identified by an algorithm as possible AF are reviewed by cardiologists. Participants with AF are seen by a general practitioner for consideration of anticoagulation. The primary outcome is stroke. Major secondary outcomes are: death, major bleeding and cardiovascular events. Follow-up will be via electronic health records for an average of 4 years. The primary analysis will be by intention-to-treat using time-to-event modelling. Results from this trial will be combined with follow-up data from the cluster-randomised pilot trial by fixed-effects meta-analysis.

**Ethics and dissemination** The London—Central National Health Service Research Ethics Committee (19/LO/1597) provided ethical approval. Dissemination will include public-friendly summaries, reports and engagement with the UK National Screening Committee.

**Trial registration number** ISRCTN72104369.

## STRENGTHS AND LIMITATIONS OF THIS STUDY

⇒ This trial is more than twice the size of previous trials of atrial fibrillation (AF) screening and has adequate power to determine whether screening reduces risk of stroke.
⇒ The power calculation has been refined based on pilot data and the results of an earlier trial which used the same AF screening device.
⇒ The screening intervention has been demonstrated by our feasibility and pilot studies to be feasible for national rollout if shown to be effective.
⇒ There is a risk of contamination in the control group due to increasing availability of personal devices that enable self-screening for AF.
⇒ Outcome data rely on electronic capture of routine data which risks incomplete ascertainment.

## INTRODUCTION

The rationale for the Screening for Atrial Fibrillation with ECG to Reduce stroke (SAFER) trial has been described previously.[1] In brief, there is insufficient evidence that the potential benefits from screening for atrial fibrillation (AF) outweigh the potential harms.[2] Recent trials have failed to demonstrate that single time point screening identifies more AF than usual care.[3–5] This is likely to be due to better AF identification within usual care than was prevalent when the Screening for Atrial Fibrillation in the Elderly (SAFE) trial demonstrated the value of single time point screening in identifying additional cases of AF in the early 2000s.[6] Therefore, interest has focused on newer technologies that enable

continuous or intermittent heart rhythm monitoring, such as hand-held ECGs, patches and implantable loop recorders.[7–9] These approaches do identify more AF than usual care, but have not been shown to reduce incidence of stroke.[7–9] Since these devices predominantly identify paroxysmal AF, it is important to determine whether such screening translates into reduced incidence of stroke, as paroxysmal AF may be associated with a lower risk of stroke than permanent AF.[10]

While the evidence base for stroke risk reduction with anticoagulation in AF is based on trials that included participants with paroxysmal AF, the new technologies diagnose people with lower AF burden than will have been typical of those with (usually symptomatic) paroxysmal AF in these trials.[11] Stroke risk in paroxysmal AF is related to AF burden,[12] so it is conceivable that people with low-burden paroxysmal AF may not benefit from anticoagulation. Indeed, this was the tentative conclusion drawn by the LOOP Study investigators who diagnosed AF in over 30% of the intervention arm of a screening trial using an implantable loop recorder.[8]

The emergence of consumer-led screening over recent years has provided further impetus to the SAFER trial.[13] Several commercially available devices are directly marketed to consumers for detection of AF.[13] The results of SAFER will also guide clinicians on the appropriate course of action in AF identified through consumer-led screening.[13]

In addition to stroke prevention, there are other benefits to treating AF with anticoagulation, including improved survival and reduced risk of myocardial infarction.[11] Indeed, the STROKESTOP screening trial reported a marginally significant reduction in a revised composite primary endpoint of stroke, systemic embolism, bleeding leading to hospitalisation and all-cause death.[9] Another potential benefit of screening for AF is to reduce risk of cognitive decline and vascular dementia.[14–17]

In terms of harm, the major concern is risk of bleeding as a result of anticoagulation of people identified as being in AF. There is clear evidence in the trials of treatment of AF with anticoagulation that benefit outweighs harm,[11] but the ratio of benefit to harm of treatment might be different for people with AF identified through screening. For example, in the LOOP trial, the 20% relative risk reduction in stroke was largely offset by the 26% relative increase in risk of major bleeding.[8] This concern is reinforced by the results of recent trials of anticoagulation in subclinical AF and atrial high-rate episodes detected as a result of implanted devices such as pacemakers, defibrillators and loop recorders (ie, not identified as a result of screening).[18 19] In the NOAH-AFNET6 trial, a non-significant 19% reduction in the primary efficacy outcome (composite of cardiovascular death, stroke and systemic embolism) was offset by a significant 31% increase in the risk of a safety outcome occurring (death from any cause or major bleeding).[18] In the ARTESIA trial, a significant 37% reduction in risk of stroke or systemic embolism was offset by a significant 36% increase in the risk of major bleeding.[19]

The aim of the SAFER trial is to determine if population screening for AF using a hand-held single-lead ECG device intermittently over a period of 3 weeks is effective and cost-effective at reducing stroke compared with usual care and to quantify other potential benefits and harms of screening. The design of the SAFER pilot trial (now successfully completed) has already been reported.[1] This protocol paper therefore focuses on changes in methods between the pilot and the main trial. The Standard Protocol Items: Recommendations for Interventional Trials checklist provides the structure for this paper.[20]

## METHODS AND ANALYSIS
### Design
SAFER is a multicentre randomised controlled trial. Randomisation is at the individual level with clustering by household (ie, if there is more than one participant from the same address, they will be allocated to the same arm). This is a change from the original intention to randomise at the level of the general practice (GP).[1] This decision was made during the internal pilot trial, when it became clear that remote delivery of the screening intervention greatly reduced the risk of contamination, so negating the value of cluster randomisation by practice. However, it was recognised that there would be a residual risk of contamination if members of the same household were in different arms of the trial. The first participant was randomised in March 2022. It is currently estimated that randomisation will finish in April 2024 and follow-up will finish in March 2027. The trial design is summarised in figure 1.

### Participants
Participant eligibility is unchanged from the pilot study, being people aged 70 years or older who are registered with participating GPs.[1] Those who are on the practice palliative care register or in a nursing or residential home are excluded, as are those already on anticoagulation therapy. Non-UK residents are excluded, as are people who have already consented to another trial that may affect participation in SAFER. People with a prior record of AF but not currently on anticoagulation are eligible as this may encourage anticoagulation use in these participants as was observed in STROKESTOP.[1] GPs are being recruited from throughout England. It is anticipated that about 195 practices will be involved.

### Recruitment
Unlike in the pilot cluster-randomised trial, where there was little gain in power from increasing sample size in each cluster, all eligible patients (as opposed to a random sample) are sent an invitation pack by their practice. This includes a consent form (see online supplemental file 1) to be returned to the trial team either by post or online.

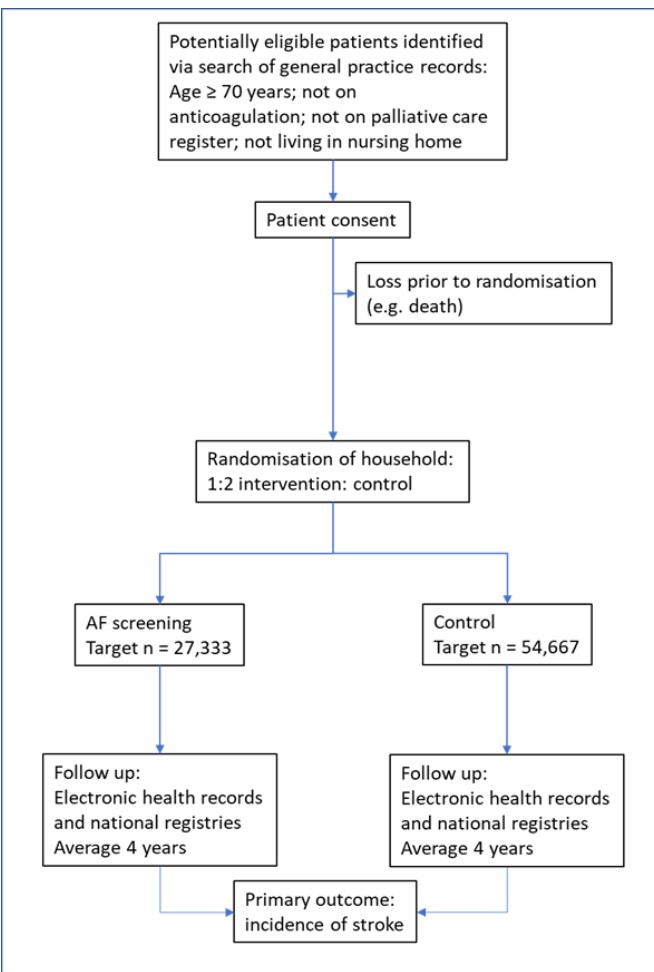

**Figure 1** Screening for Atrial Fibrillation with ECG to Reduce stroke trial schema. AF, atrial fibrillation.

## Randomisation

Randomisation is performed online at the Oxford Primary Care Clinical Trials Unit following return of consent forms, stratified by practice. Random permuted blocks ensure allocations are balanced at a ratio of 2:1 (control:intervention) in batches per practice. If there is more than one participant in the same household, they are randomised as a cluster to the same arm. In recognition that trial capacity would be limited primarily by how many participants could be screened, a 2:1 randomisation ratio was used to increase trial power for a given number of participants randomised to screening.

## Baseline data

This is unchanged from the pilot trial, includes demographics and comorbidities, and is collected from the GP electronic medical records.[1]

## Screening intervention

This is unchanged from the pilot trial.[1] In brief, participants randomised to screening will receive a further postal invitation to participate in screening. Those who accept this invitation receive a call from the trial team to arrange delivery of the single-lead ECG device (Zenicor

One, Zenicor medical systems) and instructions (written with online video available) and an offer of subsequent support by telephone on how to use it. They are asked to carry out screening four times a day for 3 weeks and take additional traces if symptomatic (eg, palpitations, dizziness). Each trace runs for 30 s. Participants transmit their recordings to a remote database using the mobile capability within the device. Each ECG is tagged with a unique participant ID number.

A proprietary algorithm (Cardiolund) analyses the ECG traces,[21] and those that show possible AF are reviewed by a cardiologist or cardiac technician. A second cardiologist performs additional review if there is uncertainty. AF is diagnosed if the rhythm is present continuously for 30 s. The screening results are returned to the practice, which notifies all participants of their results, and actively follows up those with AF or other significant diagnoses (eg, ventricular tachycardia, high-degree atrioventricular block). Participating GPs receive initial training when the practice is set up for the trial. This includes a reminder that confirmation of the diagnosis of AF with a 12-lead ECG is not required for diagnosis of paroxysmal AF.[22] They are offered further online training on the National Institute for Health and Care Excellence AF guidelines.[22] GPs are asked to provide a reason if they do not initiate anticoagulation for a participant diagnosed through screening.

## Usual care

Participants assigned to the control arm will receive usual care, which might involve single time point opportunistic screening.

## Follow-up

The target follow-up duration has been reduced from an average of 5 years (as per the pilot protocol)[1] to an average of 4 years per participant. This is to compensate for the delays imposed on the trial by COVID-19, and to lower the risk of control group contamination with risking direct marketing of AF detection devices directly to the public.[13] The revised sample size calculation (see below) takes this reduced length of follow-up into account. The Programme Steering Committee will review stroke rate in the whole trial population (ie, not by treatment arm) and may recommend modifying follow-up duration if stroke rates differ from what is expected (approximately 1% per annum).[9] Follow-up will be by electronic health records (including GP records), Hospital Episode Statistics (HES), Office for National Statistics (ONS) mortality data and national disease registries accessed via National Health Service (NHS) England and ORCHID database.[23] Participants are linked to these databases via a unique number (their NHS number). HES provides principal and secondary diagnosis codes for all hospital admissions in England. ONS mortality data include date of death, and underlying and contributory causes of death for all deaths. National disease registries provide an alternative source for stroke and myocardial infarction to HES. A

comparison of these sources suggests that data capture is more complete with combination of sources.[24]

Funding for longer-term follow-up will be sought. In particular, if AF screening is associated with reduction in dementia, the screening benefit will manifest over a longer time period.

## Outcomes

The primary outcome is stroke. This includes stroke of any severity, but excludes events only labelled as transient ischaemic attack. For the primary endpoint, ischaemic and haemorrhagic stroke events will be combined.

Secondary outcomes include: all-cause death, cardiovascular death, major adverse cardiovascular event (composite of myocardial infarction, stroke and other hospital admissions for cardiovascular disease, including heart failure), myocardial infarction, ischaemic stroke, haemorrhagic stroke, major bleeding episode (defined as requiring hospital admission), new diagnosis of dementia and new diagnosis of depression. AF detection rates and anticoagulation uptake will be reported (principal outcomes of the internal pilot trial).

Outcome ascertainment will be restricted to data available from electronic health records without event adjudication. A comparison of routine versus adjudicated follow-up in a vascular events outcome trial found that specificity of routine data was high (over 99%), and that sensitivity was over 80% if transient ischaemic attack was excluded.[25] Furthermore, there was no difference in effect size between the two sources of data.[25] The sample size calculation below takes into account the 80% sensitivity, in that it is based on observed stroke rates in a trial where the follow-up also relied on routinely available data.[9]

## Sample size

The sample size calculation has been updated to reflect the changes in trial design, the result of a recent trial of screening for AF using the Zenicor One device,[9] the interim results of the internal pilot trial and initial baseline findings from the main trial. In the STROKESTOP trial, an 8% reduction in risk of stroke was observed.[9] Due to higher uptake of screening in the intervention arm of SAFER, and the greater observed differences in AF detection rates between intervention and control as compared with STROKESTOP, a 12% relative risk reduction in stroke is now anticipated in SAFER. Assuming a household cluster size of 1.21 (from observed cluster size to date), a household intraclass correlation coefficient of 0.2[26] and a 1% annual risk of stroke in the control arm,[9] this equates to needing 82 000 participants to detect a 12% relative reduction in risk of stroke after 4 years with 90% power. Overall, the target number of participants was reduced from 126 000 to 82 000, primarily as a result of the change from being a cluster-randomised trial at the level of the practice to randomisation by household. Our experience in our feasibility and pilot studies (which will be reported separately) suggests that this number will be achievable.

## Analysis

The intention-to-treat principle will guide data analysis (outcome in all eligible randomised participants will be compared between intervention and control). All eligible randomised participants will be included in the analysis, regardless of participation in screening.

The primary analysis will be conducted separately for the cluster-randomised pilot trial and the main trial, with results then combined by fixed-effects meta-analysis. Time-to-event modelling (ie, a Cox proportional hazards model) will be used to obtain an estimate (HR) of the effect of screening on stroke risk (fatal and non-fatal), censoring other causes of death. Analysis time will be from date of randomisation.

Clustering (by practice for pilot trial participants and by household for main trial participants) will be accounted for using a robust sandwich estimator of the covariance matrix. The estimate of intervention effect will be adjusted for prespecified baseline covariates such as age and sex. Secondary outcomes will be analysed in a similar way.

For all analyses, we will test model assumptions. Should these be violated, flexible parametric survival models will be considered to model the change in HR over time.

A full statistical analysis plan will be lodged with the ISRCTN registration prior to data lock.

## Economic analysis

To determine whether screening is cost-effective from the perspective of the NHS, we will adapt an existing economic model.[27] This will incorporate data from the SAFER trial, including outcomes such as mortality and cardiovascular endpoints, to determine incremental cost per quality-adjusted life-year gained comparing screening versus no screening over a 4-year time horizon. The model parameters that do not come from the trial will be derived from updated literature reviews. We will extend the model to a lifetime horizon and consider the impact on cost-effectiveness of repeated screening at different time intervals and in different age groups.

## Management and oversight

Management and oversight is delivered through the same structure as in the pilot trial.[1] The University of Cambridge and NHS Cambridgeshire & Peterborough Integrated Care Board are co-sponsors. The trial management group meets monthly to review operational issues. The Programme Steering Committee, which has an independent chair and four independent members, provides independent oversight of the programme and acts as the Trial Steering Committee. An active risk register has been compiled in consultation with the funder and sponsors and will be monitored and updated throughout.

## Patient and public involvement

The same approach is being used as in the pilot trial.[1] In brief, we have engagement by patient and public involvement members as an investigator (Trudie Lobban, chief executive of the Atrial Fibrillation Association (AFA)) and as contributors independent of the AFA.

## Ethics and dissemination

Ethical approval has been provided by the London—Central NHS Research Ethics Committee (19/LO/1597).

In addition to peer-reviewed publications and presentation at conferences, public-friendly trial summary documents will be made available to participants at the end of the trial. Accessible reports will be generated for the UK National Screening Committee, commissioners and other decision-makers. The pilot study protocol provides further details.[1]

Requests for pseudonymised data will be directed to the trial coordinator (Andrew Dymond using SAFER@medschl.cam.ac.uk) and will be considered by the investigators, in accordance with participant consent.

**Author affiliations**
[1]Primary Care Unit, University of Cambridge, Strangeways Research Laboratory, Cambridge, UK
[2]Department of Population Health Sciences, University of Leicester, Leicester, UK
[3]THIS Labs, Cambridge, UK
[4]Liverpool Heart and Chest Hospital NHS Foundation Trust, Liverpool, UK
[5]School of Cardiovascular and Metabolic Medicine & Sciences, King's College London, London, UK
[6]Charles Perkins Centre, The University of Sydney, Sydney, New South Wales, Australia
[7]Institute of Public Health, University of Cambridge Primary Care Unit, Cambridge, UK
[8]MRC Epidemiology Unit, Cambridge, UK
[9]Department of Public Health and Primary Care, University of Cambridge Primary Care Unit, Cambridge, UK
[10]Nuffield Department of Primary Care Health Sciences, University of Oxford, Oxford, UK
[11]University of Bristol, Bristol, UK
[12]Cambridge Biomedical Campus, Cambridge, UK
[13]Danish Centre for Health Services Research, Department of Clinical Medicine, Aalborg University, Aalborg, Denmark
[14]Arrhythmia Alliance and AF Association, Stratford upon Avon, UK
[15]School of Primary Care, Population Sciences and Medical Education, University of Southampton, Southampton, UK
[16]Primary Care Unit, Department of Public Health & Primary Care, Strangeways Research Laboratory, Cambridge, UK
[17]THIS Institute, University of Cambridge, Cambridge, UK

**Acknowledgements** We would like to acknowledge the support of the Independent Programme Steering Committee: Professor Christian Mallen, University of Keele (chair); Professor Anthony Rudd, King's College London (independent member); Professor Ann Marie Swart, University of East Anglia (independent member); Professor Andy Vail, University of Manchester (independent member); Dr Bob Ward (independent lay member); and patient and public involvement representatives: Margaret Corbett, Jennifer Crockford, Trudie Lobban (founder & CEO of Atrial Fibrillation Association), Sheilah Rengert, and Dr Bob Ward.

**Contributors** JM is the guarantor and drafted the manuscript with help from RNM. KW and AD are coordinating, planning, gaining ethical approval, conducting and helping design the trial. JM, JB, NA, DE, RJ, JL, TL, ML, GYHL, BF, SJG, SS, FDRH and RJM undertook design and planning and are overseeing conduct of the trial. SH and AP as qualitative researchers contributed to the design of the intervention.

MC helped design the trial. TL is a PPI representative who has informed the design, outcomes and dissemination plan. SM and HT designed the economic evaluation and will oversee its conduct. SK designed the statistical analysis and will oversee its conduct. MS contributed to the initial work on the trial design and led statistical methods. SK contributed to revision of the trial design and will lead the development of the statistical analysis plan and oversee progress. GYHL, PC, MTM, WYD and RP conducted and refined the cardiology review process of the intervention. The SAFER author group contributed to planning and design of the trial, applying for funding and writing of the protocol for the ethical approval and have oversight of the conduct of the trial. All authors reviewed and had the option to edit the final manuscript.

**Funding** This SAFER trial is funded by the National Institute for Health and Care Research (NIHR) (Programme Grants for Applied Research Programme (reference number RP-PG-0217-20007)). SAFER is a contributor to/partner of AFFECT-EU receiving funding from the European Union's Horizon 2020 Research and Innovation Programme under grant agreement no. 847770. RNM and JL are supported by the Wellcome Trust as part of the Wellcome Trust PhD Programme for Primary Care Clinicians (grant number 203921/Z/16/Z). AP is based in The Healthcare Improvement Studies Institute (THIS Institute), University of Cambridge. THIS Institute is supported by the Health Foundation, an independent charity committed to bringing about better health and healthcare for people in the UK. FDRH acknowledges support from NIHR Applied Research Collaboration (ARC) OTV and Oxford BRC (OUT). RJM is an NIHR senior investigator and acknowledges support from NIHR ARC OTV. NA is supported by the NIHR Greater Manchester Patient Safety Research Collaboration (GM PSRC) and the NIHR ARC East Midlands (EM). RJ is an NIHR-funded academic clinical lecturer. The University of Cambridge has received salary support in respect of SJG from the NHS in the East of England through the Clinical Academic Reserve. BF received funding from the Medical Research Future Fund International Clinical Trial Collaboration Grant to perform SAFER-AUS as part of SAFER, and an NSW Health Senior Researcher Cardiovascular Grant for work in AF. GYHL is an NIHR senior investigator and co-principal investigator of the AFFIRMO Project on multimorbidity in AF, which has received funding from the European Union's Horizon 2020 Research and Innovation Programme under grant agreement no. 899871.

**Disclaimer** All the funders and sponsors had no involvement in the development of this protocol and will have no involvement in any aspect of the trial itself. The views expressed are those of the author(s) and not necessarily those of the NHS, the Wellcome Trust, the NIHR or the UK Department of Health and Social Care.

**Competing interests** JM has performed consultancy work for BMS/Pfizer and Omron. FDRH reports occasional consultancy for BMS/Pfizer, Bayer and BI over the past 5 years. NA is a member of the UK National Screening Committee. MC and MS are employed by AstraZeneca, but at the time of involvement with the trial were employed by universities (King's College London and University of Leicester, respectively), for which they still hold honorary contracts. RJM's employer, the University of Oxford, receives consultancy and licensing payments from Omron and Sensyne for BP telemonitoring interventions. GYHL is a consultant and speaker for BMS/Pfizer, Boehringer Ingelheim, Daiichi-Sankyo and Anthos. No fees are received personally. SJG has received honoraria from AstraZeneca for lectures at postgraduate educational meetings for primary care teams about type 2 diabetes. BF has received speaker fees, honoraria and non-financial support from BMS and Pfizer Alliance; grants to the institution for investigator-initiated studies from BMS and Pfizer Alliance; and loan devices for investigator-initiated studies from Alivecor, all were unrelated to the present trial but related to screening for AF.

**Patient and public involvement** Patients and/or the public were involved in the design, or conduct, or reporting, or dissemination plans of this research. Refer to the Methods section for further details.

**Patient consent for publication** Not applicable.

**Provenance and peer review** Not commissioned; externally peer reviewed.

**ORCID iDs**
Rakesh N Modi http://orcid.org/0000-0001-9651-6690
Natalie Armstrong http://orcid.org/0000-0003-4046-0119
Gregory Y H Lip http://orcid.org/0000-0002-7566-1626
Mark Lown http://orcid.org/0000-0001-8309-568X
Richard J McManus http://orcid.org/0000-0003-3638-028X
Stephen Morris http://orcid.org/0000-0002-5828-3563
Stephen Sutton http://orcid.org/0000-0003-1610-0404
Kate Williams http://orcid.org/0000-0002-6188-9363

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
