## [Reviewer comments · BMJ Open]

ARTICLE DETAILS

TITLE (PROVISIONAL)	Randomised controlled trial of population screening for atrial fibrillation in people aged 70 years and over to reduce stroke: protocol for the SAFER trial.
AUTHORS	Mant, Jonathan; Modi, Rakesh; Dymond, Andrew; Armstrong, Natalie; Burt, Jenni; Calvert, Peter; Cowie, Martin; Ding, Wern; Edwards, Duncan; Freedman, Ben; Griffin, Simon; Hoare, Sarah; Hobbs, Richard; Johnson, Rachel; Kaptoge, Stephen; Lip, Gregory; Lobban, Trudie; Lown, Mark; Lund, Jenny; McManus, Richard; Mills, Mark; Morris, Stephen; Powell, Alison; Proietti, Riccardo; Sutton, Stephen; Sweeting, Mike; Thom, Howard; Williams, Kate; SAFER Authorship Group, The

VERSION 1 – REVIEW

REVIEWER	Gruwez, Henri Ziekenhuis Oost-Limburg, Cardiology
REVIEW RETURNED	03-Dec-2023

GENERAL COMMENTS	The authors very nicely describe the protocol of an atrial fibrillation (AF) screening study. I only have very minor comments and look forward to the study results. - I particularly enjoyed reading the introduction. There have been several large AF screening trials that demonstrated the difficulty to identify the population, AF burden and AF type in which the benefit of oral anticoagulation (OAC) may outweigh the bleeding risk. The authors have very well summarized current knowledge. It may be useful to tough upon the findings of the recent NOAF-AFNET 7 trials and ARTESIA trial, specifically because these the stroke risk of subclinical AF will strongly determine the event rate in the SAFER trial.- I want to congratulate the authors on the ambitious endpoints. They are well chosen for clinical relevance and impact of AF screening. Also, the trial design will be easily interpretable and generalizable towards real world deployment. Also, the secondary endpoints are relevant. Will changes in AF management (other than OAC) also be measured? It might be useful to learn if more changes in AF management (antiarrhythmic drugs (AAD) or direct cardioversions (DCV)) lead to fewer events of the primary endpoint (also interesting if more AAD and DCV does not lead to fewer events).- The primary endpoint, stroke, also excludes TIA. I applaud this choice. However, this will strongly reduce the number of events. Maybe consider accepting TIA as an event if confirmed on MRI. Or for example, if it occurs in conjunction with a thrombus in the left atrial appendix. Maybe, as a secondary analysis...- Page 9, line 20-22: a verb seems missing here.
--

	- Please specify: page 11 line 27-31: 'The programme steering committee will review stroke rate in the whole study population (i.e., not by treatment arm), and may recommend modifying follow up duration if stroke rates differ from what is expected. - Can the authors explain why they chose for a 2:1 randomization (as opposed to 1:1)?
--	--

REVIEWER	Bauer, Axel University of Innsbruck, Dpt. of Cardiology
REVIEW RETURNED	04-Dec-2023

GENERAL COMMENTS	The article describes an updated protocol of the SAFER study, as significant changes have been made since start of enrollment and after completion of the pilot study. In the SAFER study, 82,000 people aged 70 years or older are randomized in a 1:2 ratio to either atrial fibrillation screening with a handheld ECG or to a control group. Participants diagnosed with atrial fibrillation will be treated by trained general practitioners involved in the study. Participants will be followed up for 4 years to determine the primary endpoint of ischemic or hemorrhagic stroke. The SAFER study is an extremely important study in the field of prevention that will have a significant impact on future strategies for the early detection of atrial fibrillation. The changes to the protocol compared to the previous version are well thought out, well argued and balanced and will not compromise the conclusion or impact of the study. I have only minor comments or questions:  - Did the change in randomization (household vs. cluster) have an impact on the power calculation? - Was the Zenicor device per-programmed to ensure identification of the individuals? - Are online trainings on guidelines for GPs mandatory? Do you determine adherence to the guidelines in the study? - Did you estimate the accuracy of follow-up information from the electronic systems? What is your estimation? Will medical reports be requested for potential endpoints? - Recruitment in the study is of crucial importance. What was your experience in the pilot study (e.g. percentage of invited and enrolled participants) and were there differences between the invited and enrolled participants?
--

REVIEWER	Brunetti, Enrico University of Turin
REVIEW RETURNED	07-Dec-2023

GENERAL COMMENTS	This manuscript presents the protocol of an ongoing randomized clinical trial aiming to evaluate the impact of widespread atrial fibrillation screening in the community-dwelling UK population of subjects aged 70 years and older by use of a handheld single lead ECG 4 times daily for a week on the incidence of the combined of ischemic and hemorrhagic stroke. This trial follows the results of a pilot trial, whose protocol had already been published in the BMJ open. The research question is relevant and up-to-date, and the protocol proposed is suitable to respond to this research question. However I believe that some points need minor clarifications:
---

	 - Have the Authors thought about any prespecified subanalyses (e.g. by age strata, gender, etc)? If so, please elaborate them in the manuscript, to avoid having them labelled as post-hoc analyses. - Does the study protocol include some kind of evaluation on the impact of AF screening and detection on indirect therapeutic measures (e.g. cardiovascular risk factor management, rhythm-control strategies...)? - I believe that the follow-up time is too short to make the dementia incidence outcome a realistic one, also given the huge lag time between brain damage and cognitive impairment development, and between the cognitive impairment development and actual dementia diagnosis - The design section says that randomization will finish in January 2024, and follow-up in March 2027. However, the protocol states that follow-up time will be of 4 years. Patients enrolled in Jan 2024 will have therefore a follow up of just 2 years and 2 months? Please clarify this matter. - At page 10 lines 3-5 you state that "People with a prior record of AF but not currently on anticoagulation are eligible". I cannot see the point of enrolling in a screening program patients for whom the index condition is already known: what is the expected benefit of the intervention for this population (both from a practical and ethical point of view)? - Allocation (page 10 line 31): why did you decide for an allocation ratio of 2 controls vs 1 intervention? Perhaps the reader would benefit in knowing the reasons behind this decision. - Screening intervention (page 11 line 48): I am not sure I got the procedure of patient enrolment and allocation right. You state "In brief, participants randomised to screening will receive a postal invitation to participate." I have understood that all patients in the participating general practices will receive a postal invitation to participate in the study, and consenting individuals (responders) will then be randomised to intervention vs control. Am I right? Or there is a further step in which patients randomised to the intervention/screening device, will be contacted to accept/not accept the intervention? - Screening intervention (page 11): only patients with a suspect AF diagnosis will be returned the results of the screening, right? When the one lead ECG will give an AF diagnosis, will the diagnosis be confirmed by conventional 12-lead ECG? This is for example mandatory in my country (Italy) in order to prescribe an oral anticoagulation with the National Health Service. - Follow-up (page 11): since the BMJ Open readership is international, I believe that some perspective on how the electronic health records and national databases and registries are organised would be useful to fully understand outcome ascertainment. - Outcomes: will be ischemic and hemorrhagic strokes also be adjudicated and evaluated separately? - I believe that not every listed Author of the manuscript has been acknowledged in the author contributions section. This is important for the ICMJE Authorship criteria. Moreover, I see that two Authors (MC and MS) are both affiliated to a major Company (Astrazeneca), but no further details about their involvement in the study design, planning and conduct are given. Please provide more details.
--	--

	Also, please note that the last sentence of the introduction (page 9 lines 20-22) is incomplete "The SPIRIT checklist when writing this paper."
--	---

VERSION 1 – AUTHOR RESPONSE

Reviewer 1

- I particularly enjoyed reading the introduction. There have been several large AF screening trials that demonstrated the difficulty to identify the population, AF burden and AF type in which the benefit of oral anticoagulation (OAC) may outweigh the bleeding risk. The authors have very well summarized current knowledge. It may be useful to tough upon the findings of the recent NOAH-AFNET 7 trials and ARTESIA trial, specifically because these the stroke risk of subclinical AF will strongly determine the event rate in the SAFER trial.

Thank you. We have added sentences referring to these two studies in the introduction: **This concern is reinforced by the results of recent trials of anticoagulation in sub-clinical atrial fibrillation and atrial high rate episodes detected as a result of implanted devices such as pacemakers, defibrillators and loop recorders (i.e. not identified as a result of screening). In the NOAH-AFNET6 trial, a non-significant 19% reduction in the primary efficacy outcome (composite of cardiovascular death, stroke and systemic embolism) was offset by a significant 31% increase in the risk of a safety outcome occurring (death from any cause or major bleeding). In the ARTESIA trial, a significant 37% reduction in risk of stroke or systemic embolism was offset by a significant 36% increase in the risk of major bleeding.**

- I want to congratulate the authors on the ambitious endpoints. They are well chosen for clinical relevance and impact of AF screening. Also, the trial design will be easily interpretable and generalizable towards real world deployment. Also, the secondary endpoints are relevant. Will changes in AF management (other than OAC) also be measured? It might be useful to learn if more changes in AF management (antiarrhythmic drugs (AAD) or direct cardioversions (DCV)) lead to fewer events of the primary endpoint (also interesting if more AAD and DCV does not lead to fewer events).

Thank you. We are not measuring other changes in AF management. While we recognise the potential value of such additional data, our strategy with regard to primary care data extractions are to keep them as simple as possible. We do not anticipate major differences in AADs or DCV between the two arms, as the additional AF cases identified in the intervention arm are likely to be asymptomatic. No changes made to manuscript.

- The primary endpoint, stroke, also excludes TIA. I applaud this choice. However, this will strongly reduce the number of events. Maybe consider accepting TIA as an event if confirmed on MRI. Or for example, if it occurs in conjunction with a thrombus in the left atrial appendix. Maybe, as a secondary analysis...

Since we are reliant on electronic data collection for our outcome detection, TIAs are likely to be missed. This links to a question from reviewer 2. We have added text to the paper: **A comparison of routine versus adjudicated follow up in a vascular events outcome trial found that specificity of routine data were high (over 99%), and that sensitivity was over 80% if transient ischaemic attack was excluded.**

- Page 9, line 20-22: a verb seems missing here.

This has been corrected – see final comment under editorial comments above.

- Please specify: page 11 line 27-31: 'The programme steering committee will review stroke rate in the whole study population (i.e., not by treatment arm), and may recommend modifying follow up duration if stroke rates differ from what is expected.

We have added in parentheses: **(approximately 1% per annum)**

- Can the authors explain why they chose for a 2:1 randomization (as opposed to 1:1)?

We have added the following: '**In recognition that trial capacity would be limited primarily by how many participants could be screened, a 2:1 randomisation ratio was used to increase study power for a given number of participants randomised to screening**'.

Reviewer 2

- Did the change in randomization (household vs. cluster) have an impact on the power calculation?

We have added the following: **Overall, the target number of participants was reduced from 126,000 to 82,000, primarily as a result of the change from being a cluster randomised trial at the level of the practice to randomisation by household.**

- Was the Zenicor device pre-programmed to ensure identification of the individuals?

It was pre-programmed to the extent that: 'A proprietary algorithm (Cardiolund) analyses the ECG traces' (Start of second paragraph under 'Screening Intervention'). To ensure correct identification of individuals, each ECG is tagged with a unique participant ID number. We have added the following to the manuscript: '**Each ECG is tagged with a unique participant ID number**'.

- Are online trainings on guidelines for GPs mandatory? Do you determine adherence to the guidelines in the study?

No. The online training is optional. But there is mandatory training when the practice is set up. We monitor whether the online training is taken up, and adherence to uptake of anticoagulation (one of our secondary outcomes). If anticoagulation is not initiated following a diagnosis of AF, GPs are asked to provide a reason. We have amended the text (the expansion is partly to respond to a query from reviewer 3). : Participating GPs receive **initial training when the Practice is set up for the trial. This includes a reminder that confirmation of the diagnosis of AF with a 12 lead ECG is not required for diagnosis of paroxysmal AF. -They are offered further** on line training...

- Did you estimate the accuracy of follow-up information from the electronic systems? What is your estimation? Will medical reports be requested for potential endpoints?

To clarify our approach, we have added the following to the section on outcomes: **Outcome ascertainment will be restricted to data available from electronic health records without event adjudication. A comparison of routine versus adjudicated follow up in a vascular events outcome trial found that specificity of routine data was high (over 99%), and that sensitivity**

was over 80% if transient ischaemic attack was excluded. Furthermore, there was no difference in effect size if the analysis excluded the adjudicated direct follow up data. The sample size calculation below takes into account the 80% sensitivity, in that it is based on observed stroke rates in a trial where the follow up also relied on routinely available data.

- Recruitment in the study is of crucial importance. What was your experience in the pilot study (e.g. percentage of invited and enrolled participants) and were there differences between the invited and enrolled participants?

We will be reporting separately results of our feasibility and pilot studies. We have added the following: **Overall, the target number of participants was reduced from 126,000 to 82,000, primarily as a result of the change from being a cluster randomised trial at the level of the practice to randomisation by household. Our experience in our feasibility and pilot studies (which will be reported separately) suggest that this number will be achievable.**

Reviewer 3

- Have the Authors thought about any prespecified subanalyses (e.g. by age strata, gender, etc)? If so, please elaborate them in the manuscript, to avoid having them labelled as post-hoc analyses.

We have clarified in the manuscript that the Statistical Analysis Plan has not yet been finalised: **'A full statistical analysis plan will be lodged with the ISRCTN registration prior to data lock.'**

- Does the study protocol include some kind of evaluation on the impact of AF screening and detection on indirect therapeutic measures (e.g. cardiovascular risk factor management, rhythm-control strategies...)?

No. See response to reviewer 1, who raised a similar question.

- I believe that the follow-up times is too short to make the dementia incidence outcome a realistic one, also given the huge lag time between brain damage and cognitive impairment development, and between the cognitive impairment development and actual dementia diagnosis

You may well be right. We have added an acknowledgement of this limitation, and note that longer term follow up may provide more evidence: **'Funding for longer term follow up will be sought. In particular, if AF screening is associated with reduction in dementia, then this reduction might be expected to manifest itself over a longer time period.'**

- The design section says that randomization will finish in January 2024, and follow-up in March 2027. However, the protocol states that follow-up time will be of 4 years. Patients enrolled in Jan 2024 will have therefore a follow up of just 2 years and 2 months? Please clarify this matter.

The protocol states that the average follow up will be of four years. To re-emphasise this, we have inserted an average of: The target follow up duration has been reduced from an average of five years (as per the pilot protocol) to **an average of** four years per participant

- At page 10 lines 3-5 you state that "People with a prior record of AF but not currently on anticoagulation are eligible". I cannot see the point of enrolling in a screening program patients for whom the index condition is already known: what is the expected benefit of the intervention for this population (both from a practical and ethical point of view)?

This was justified in our earlier protocol paper for the pilot study that was published in BMJ Open, which is why we did not repeat the justification here. In that protocol we state: 'participants with an existing diagnosis of AF on the practice electronic AF register but who are not being prescribed anticoagulation are included because screening these participants for AF may encourage anti-coagulation use'. We give reference to the STROKESTOP study, which also included such individuals. We have added a shorter version of this to the paper, and cross referenced the protocol paper: **'People with a prior record of AF but not currently on anticoagulation are eligible as this may encourage anticoagulation use in these participants as was observed in STROKESTOP'**

- Allocation (page 10 line 31): why did you decide for an allocation ratio of 2 controls vs 1 intervention? Perhaps the reader would benefit in knowing the reasons behind this decision.

See response to reviewer 1.

- Screening intervention (page 11 line 48): I am not sure I got the procedure of patient enrolment and allocation right. You state "In brief, participants randomised to screening will receive a postal invitation to participate." I have understood that all patients in the participating general practices will receive a postal invitation to participate in the study, and consenting individuals (responders) will then be randomised to intervention vs control. Am I right? Or there is a further step in which patients randomised to the intervention/screening device, will be contacted to accept/not accept the intervention?

All participants are invited to take part in the trial. Participants randomised to the intervention are then further invited to take part in screening. We have added to the text: In brief, participants randomised to screening will receive a **further** postal invitation to participate **in screening**.

- Screening intervention (page 11): only patients with a suspect AF diagnosis will be returned the results of the screening, right?

No – the practice informs all patients. We have amended the text: 'The results are returned to the practice, which notifies **all** participants of the results'

-When the one lead ECG will give an AF diagnosis, will the diagnosis be confirmed by conventional 12-lead ECG? This is for example mandatory in my country (Italy) in order to prescribe an oral anticoagulation with the National Health Service.

No – since this would result in most cases of paroxysmal AF detected by screening not being treated. We have expanded our description of the GP training to make this point (see above for response to reviewer 2).

- Follow-up (page 11): since the BMJ Open readership is international, I believe that some perspective on how the electronic health records and national databases and registries are organised would be useful to fully understand outcome ascertainment.

We have added the following: **Participants are linked to these databases via a unique number (their NHS number). HES provides principal and secondary diagnosis codes for all hospital admissions in England. ONS mortality data includes date of death, and underlying and contributory causes of death for all deaths. National disease registries provide an alternative source for stroke and myocardial infarction to HES. A comparison of these sources suggests that data capture is more complete when both sources are used.**

- Outcomes: will be ischemic and hemorrhagic strokes also be adjudicated and evaluated separately?

We have added a paragraph under outcomes to clarify and explain why adjudication will not be performed. Yes, ischaemic and haemorrhagic stroke will be evaluated separately. This has been clarified under the list of secondary outcomes: 'Secondary outcomes include:**ischaemic stroke; haemorrhagic stroke**;...

- I believe that not every listed Author of the manuscript has been acknowledged in the author contributions section. This is important for the ICMJE Authorship criteria.

This has been corrected.

-Moreover, I see that two Authors (MC and MS) are both affiliated to a major Company (Astrazeneca), but no further details about their involvement in the study design, planning and conduct are given. Please provide more details.

This has been clarified both in the contributions, and in the competing interest statement: 'MRC and MS are employed by Astrazeneca PLC, **but at the time of involvement with the study were employed by Universities (Kings College London and University of Leicester respectively), for which they still hold honorary contracts.**' To avoid confusion, we have changed the attribution of MS to University of Leicester, rather than Astra Zeneca PLC.

VERSION 2 – REVIEW

REVIEWER	Gruwez, Henri Ziekenhuis Oost-Limburg, Cardiology
REVIEW RETURNED	06-Feb-2024

GENERAL COMMENTS	None.
-------

REVIEWER	Bauer, Axel University of Innsbruck, Dpt. of Cardiology
REVIEW RETURNED	02-Feb-2024

GENERAL COMMENTS	The authors appropriately addressed all the comments and revised the MS.
--

REVIEWER	Brunetti , Enrico University of Turin
REVIEW RETURNED	27-Jan-2024

GENERAL COMMENTS	The Authors have efficiently addressed all points that have been raised by Reviewers. I have nothing more to comment, if not to complimentothe Authors
--